# Pulmonary Vein Enlargement as an Independent Predictor for New-Onset Atrial Fibrillation

**DOI:** 10.3390/jcm9020401

**Published:** 2020-02-02

**Authors:** Sunwon Kim, Yong-Hyun Kim, Seung-Hwa Lee, Jin-Seok Kim

**Affiliations:** 1Division of Cardiology, Department of Internal Medicine, Korea University Ansan Hospital, Ansan-si, Gyeonggi-do 15355, Korea; sunwon11@hanmail.net (S.K.); xkyhx@hanmail.net (Y.-H.K.); 2Department of Radiology, Korea University Ansan Hospital, Ansan-si, Gyeonggi-do 15355, Korea; yaaong@hitel.net

**Keywords:** pulmonary vein, computed tomography, atrial fibrillation, diastolic function

## Abstract

Pulmonary vein (PV) enlargement is associated with atrial fibrillation (AF). However, the predictive value of PV volume for new-onset AF has not been determined. We retrospectively assessed and enrolled non-AF subjects who underwent echocardiography and cardiac CT angiography (CCTA) around the same time and evaluated the development of AF longitudinally. PV volume was assessed by estimating the three-dimensional CCTA-derived mid-diastolic PV volume from the ostium to tertiary branches. Overall, 1105 subjects were enrolled. Among them, 29 developed AF during a mean follow-up of 4.28 ± 3.08 years after baseline CCTA and echocardiography. The AF group had a higher proportion of older aged subjects, a higher ratio of early mitral flow velocity (E) to early mitral annular tissue velocity (Em), higher Em, and larger left atrial (LAVI) and PV (PVVI) volume indices. PVVI was independently associated with male sex, left ventricular dimension, E/Em and LAVI. AF incidence increased markedly across each baseline PVVI tertile (2.2%, 5.1%, and 10.8%). In the multivariate Cox model, increased PVVI was independently associated with new-onset AF (hazard ratio (HR) = 5.401, 4.931–6.193, *p* = 0.007). Based on the analysis of multimodal cardiac imaging, our results provide mechanistic insights into PV remodeling and its potential role as a link between diastolic dysfunction and developing AF.

## 1. Introduction

Atrial fibrillation (AF) is the most common sustained cardiac arrhythmia in the general population and is associated with an increased risk of morbidity and mortality [1]. The established risk factors for AF include advanced age, hypertension, diabetes, and cardiovascular disease [2]. Left atrial (LA) enlargement is also a predictive factor for developing AF and is associated with hemodynamic factors such as noncompliant left ventricle and elevated LA pressure [3].

In AF arrhythmogenesis, the pulmonary veins (PVs) play an essential role as a trigger or driver of AF [4,5,6]. Prior studies using cardiac imaging modalities demonstrated that AF patients had significantly enlarged PVs compared to controls [7,8]. This finding suggests that structural alteration of the PVs is related to AF development. The PVs and left atrium are anatomically continuous structures without barrier of blood flow; therefore, they are exposed to and share the same hemodynamic influences. Thus, primary causative factors for increased LA pressure may be involved in the structural remodeling of PVs. However, unlike LA enlargement, the prognostic value of enlarged PVs for new-onset AF is unknown. Furthermore, there is a lack of research evaluating the hemodynamic factors underlying the structural remodeling of PVs.

In this study, we hypothesized that increased PV volume could be a predictive risk factor for AF. To test our hypothesis, we analyzed imaging data and the clinical course of patients without atrial arrhythmias who underwent both echocardiography and cardiac computed tomography angiography (CCTA). Subsequently, we assessed the following data: (1) the clinical and echocardiographic determinants of the PV volume and (2) the utility of PV volume for the prediction of new-onset AF.

## 2. Materials and Methods

### 2.1. Study Population and Protocol

We retrospectively recruited 1365 adult subjects who underwent electrocardiography (ECG), CCTA, and transthoracic echocardiography within a 2-week interval between January 2002 and December 2012 at the Korea University Ansan Hospital in Ansan-si, Korea. As CCTA in our hospital was installed in 2002, we defined the study period from January 2002. Also, we set the study period of interest (2002–2012) to ensure a statistically significant volume of study subjects. The exclusion criteria included history or clinical evidence of paroxysmal or permanent AF. In addition, patients with a history of cardiomyopathy, pericardial disease, chronic kidney disease (serum creatinine ≥2 mg/dL), more than moderate valvular heart disease, congenital heart abnormality, chronic obstructive pulmonary diseases, pulmonary vascular disease or previous cardiac surgery were excluded. We defined the follow-up period as from the date of the CCTA scanning to the day of the last follow-up or to December 2017, to ensure the clinically meaningful observation time. Data regarding demographics, medical history, and clinical course after undergoing CCTA were collected. All of the serials of ECG of each subject were reviewed longitudinally by two cardiologists using the MUSE Cardiology Information System (GE Healthcare, Milwaukee, WI, USA). New-onset AF was defined as AF lasting more than 30 s detected on a 12-lead ECG or 24 h Holter monitoring during the clinical follow-up in patients with no history of paroxysmal or persistent AF or atrial flutter. The case was considered censored if (1) new-onset AF was documented or (2) the participant was lost to follow-up. The differences in baseline characteristics between patients who did and those who did not develop AF after baseline CCTA and echocardiography were assessed. Written informed consent for CCTA imaging was obtained from all subjects, and the study protocol was approved by our institutional review board (2019AS0011).

### 2.2. Imaging Acquisition and Analysis

All CCTA images were acquired using the QXi LightSpeed scanner (General Electric Medical Systems, Milwaukee, WI, USA). Cardiac cycle-gated CT scanning was performed 20–25 s after radiocontrast administration at suspended full-inspiration from the upper portion of the left diaphragm to the aortic arch with collimation 2.5 mm (table speed of 15 mm per rotation, rotation per 0.8 s with kVp = 120, mA = 230–350 depending on body habitus). Three-dimensional (3D) PV images were provided and analyzed using a dedicated image analysis software (Vitrea 2; Vital Images, Minneapolis, MN, USA). The PV ostium was defined as the point of reflection of the parietal pericardium [8]. The PV volume was determined as the mid-diastolic volume from the PV ostium to the tertiary branches of each venous tree and was calculated automatically using the embedded function of 3D image analysis software (Figure 1). The PV volume index (PVVI) indicated PV volume divided by the body surface area (BSA). CCTA specialists assessed the luminal diameter stenosis of each epicardial coronary artery segment of >2 mm in diameter. The strategy of stenosis quantification was at the investigators’ discretion. The presence of luminal stenosis ≥50% in one or more major epicardial coronary artery was considered to indicate significant coronary artery disease (CAD).

Transthoracic echocardiography was performed using commercial systems (Vivid7, GE, Vingmed Ultrasound, Horton, Norway) and reported by the American Society of Echocardiography-accredited echocardiographers. The LA volume was measured at the end-systolic phase using biplane Simpson’s method and was divided by BSA to calculate the LAVI. The E/Em ratio was calculated by dividing early diastolic transmitral flow velocity (E) by septal mitral annular tissue velocity (Em) [9]. LA enlargement was defined as LAVI ≥34 mL/m^2^ [10,11].

### 2.3. Statistical Analysis

The continuous variables were expressed as mean ± standard deviation (SD) and compared using the Student’s *t*-test or Mann-Whitney U test for non-normally distributed data. The categorical variables were reported as count and percentages and compared using the chi-square test or Fisher’s exact test as appropriate. A Kaplan-Meier analysis and log-rank test were used to compare the cumulative incidences of new-onset AF between three groups according to the baseline PVVI tertile. The determinants of PVVI were evaluated using a multiple linear regression analysis incorporating relevant clinical, demographic, and echocardiographic parameters in the model. To evaluate the impact of cardiac hemodynamics on LA and PV structures, we plotted and analyzed the LAVI and PVVI values according to E/Em, as an echocardiographic surrogate of increased left ventricular (LV) filling pressure and LA pressure overload. The best-fit curves were determined based on the R-squared statistic. Correlation coefficient analysis was performed using the Pearson or Spearman rank correlation test, depending on the normality of the distribution. The receiver-operating characteristic (ROC) curve analysis was utilized to determine the predictive performance and optimal cut-off value of the volume indices (the best Youden index = sensitivity + specificity – 1). A multivariate logistic regression analysis was performed to determine the predictors of new-onset AF. All of the potential parameters suggestive of an association with newly developed AF by univariate analysis (*p* < 0.1) and the conventional AF risk factors were included in a stepwise regression analysis as the variable-selection process; the results were reported as hazard ratios (HRs) and 95% confidence intervals (CIs). All data were analyzed using SPSS (version 20.0; IBM SPSS Statistics, IBM Corporation, Chicago, IL, USA) and GraphPad Prism (version 8.0; GraphPad Software, San Diego, CA, USA). A two-sided *p* value of <0.05 indicated statistical significance.

## 3. Results

### 3.1. Baseline Characteristics

Among the 1,365 patients initially screened, a total of 1,105 patients (mean age 58.2 ± 13.4 years; 52.9% male) met all the study criteria and were enrolled in the study (Figure 2). Overall, the population had a mild to moderate cardiovascular risk profile. The most common reason for CCTA was to evaluate chest discomfort (632 patients, 57.2%) and the second was for a general check-up (389 patients, 35.2%). The mean time-intervals across repeated ECG tests during the clinical follow-up was 10.6 months. Demographic and baseline characteristics of the study population are provided in Table 1.

### 3.2. New-onset Atrial Fibrillation and PV Enlargement

There were 29 cases of ECG-confirmed new-onset AF during a mean follow-up of 4.28 ± 3.08 years after baseline CCTA and echocardiography. Patients with new-onset AF were older than those without AF. Furthermore, patients with AF were more likely to have higher early mitral annular tissue velocity (Em), higher E/Em ratio, and larger LAVI and PVVI than those without AF (Table 1).

The incidence of new-onset AF markedly increased with baseline PVVI tertiles (tertile-1: 8.53–20.08, tertile-2: 20.08–23.8, tertile-3: 23.8–42.42 mL/m^2^). The rates per 1000 person-years across tertiles were 2.2%, 5.1%, and 10.8%, respectively (Figure 3A). A larger PVVI was associated with an increased risk of AF. Baseline PVVI tertile-3 was associated with a 4.7 times higher risk of developing AF than tertile-1 in multivariate analysis incorporating conventional AF risk factors (HR: 4.70, 95% CI: 4.07–5.33, *p* = 0.015, Figure 3B).

### 3.3. Clinical and Echocardiographic Determinants of PV Enlargement

PV volume was positively correlated with anthropometric characteristics, such as height (r = 0.388, *p* < 0.001), weight (r = 0.452, *p* < 0.001), BSA (r = 0.452, *p* < 0.001), and BMI (r = 0.260, *p* < 0.001). Among clinical and echocardiography-derived variables, PVVI was significantly associated with age, sex, history of hypertension, LV mass index, LV end-diastolic dimension (LVEDD), Em, E/Em, pulmonary arterial pressure, and LAVI. In multiple regression analysis, the independent determinants of increased PVVI were male sex, LVEDD, E/Em, and LAVI (Table 2).

### 3.4. Relationships between LAVI, PVVI and E/Em 

Both LAVI and PVVI were positively correlated with E/Em (r = 0.36, *p* < 0.0001; and r = 0.18, *p* < 0.0001, respectively; Figure 4A,B). Overall, LAVI showed a linear relationship with E/Em, while PVVI showed a small curvilinear association with E/Em. Initially, in the curve comparison based on normalized LAVI and PVVI values, the E/Em-dependent change pattern of PVVI was likely to be more gradual than that of LAVI. However, the E/Em-dependent PVVI showed a significant increase after a significant increase in E/Em (Figure 4C).

### 3.5. PVVI, LAVI and the Risk of New-Onset Atrial Fibrillation

The ROC curve revealed that both baseline PVVI and LAVI, with a cut-off value of >21.8 mL/m^2^ and >32 mL/m^2^, respectively, are potential predictors of developing AF. In the comparison of ROC curves, the predictive performance of PVVI on new-onset AF was higher than that of LAVI (area under the curve (AUC) of 0.696, *p* < 0.0001 vs. 0.651, *p* = 0.013, respectively; Figure 5).

In univariate analysis, after stratifying the PVVI and LAVI variables by cut-off values determined using ROC analysis, age, LV mass index, the E/Em ratio, and both the stratified LAVI and PVVI were associated with a higher probability of new-onset AF (Table 3). However, multivariate analysis incorporating baseline covariates and traditional AF risk factors demonstrated that only PVVI (HR = 5.401; 95% CI: 4.931–6.193, *p* = 0.007) was independently associated with developing AF (Table 3). The predictive power of LAVI for new-onset AF did not increase (HR = 1.463; 95% CI: 0.943–1.983, *p* = 0.465), even when a different LAVI criterion (LAVI >34 mL/m^2^; generally accepted definition of “increased LAVI”) was applied to the multivariate regression model.

## 4. Discussion

Prior studies that reported the association between PV enlargement and AF were mainly based on a cross-sectional analysis performed on a small population; furthermore, no follow-up data were provided [7,8]. Therefore, the predictive value of enlarged PV for developing AF has remained undetermined. Although the present study was a retrospective analysis, it was performed on a large population without significant cardiovascular disease and cardiac arrhythmia at baseline, in which patients’ clinical course was reviewed longitudinally for over 4 years. To the best of our knowledge, this study is the first to demonstrate the prognostic significance of enlarged PV for the development of AF. The main findings of this study are the following: (1) independent determinants of increased PVVI include male sex, LVEDD, E/Em, and LAVI; (2) PVVI appears to increase exponentially with the increase in E/Em; (3) the incidence of AF increases markedly as PVVI increases; and (4) the increased PVVI is significantly associated with new-onset AF, independent of other traditional risk factors.

### 4.1. Enlarged Pulmonary Vein as a Primary Substrate for Atrial Fibrillation

Enlargement of cardiac chambers and consequent myocardial stretching play a crucial role in the development of arrhythmias in various clinical conditions [12]. This stretch-induced change in electrophysiological properties, so-called mechano-electric feedback, was initially investigated at the ventricular level and has also been well-studied at the atrial level [13]. Not surprisingly, similar stretch-related arrhythmic vulnerability has been verified in the cardiomyocytes within the PV. This evidence includes in vitro work on PV isolates and data from ex-situ animal hearts in a volume overload condition [14,15,16,17]. In humans, acute volume loading-mediated mechanical stretch resulted in significant conduction slowing across the PV-LA junction, which promoted a favorable electrophysiological milieu for AF initiation [18]. In the context of such evidence, the present study provided clinical evidence supporting stretch-activated PV arrhythmogenesis for AF development. 

Enlarged PV could be involved in AF progression, as well as in the initiation of AF. In hypertensive patients with paroxysmal AF, augmented PV backflow leading to cyclic stretching of PV musculature could be a predictor of future AF perpetuation [19]. Also, patients with large PVs could be at a higher risk of recurrent AF after successful radiofrequency catheter ablation than those without [20]. These findings suggest that stretched PV, and consequent PV enlargement, can play a crucial role in AF arrhythmogenesis. 

### 4.2. Pulmonary Vein: a Plausible Crosslink between LV Diastolic Dysfunction and AF

Theoretically, in the absence of an effective anatomical barrier between the LA and PV, all hemodynamic conditions leading to elevated LA pressure can affect the structural remodeling of the PV. The LV diastolic dysfunction and its severity have been reported as independent predictors of new-onset AF; however, the pathophysiological link between LV diastolic dysfunction and AF development is not fully understood [21].

Reduced LA compliance and elevated LA pressure may influence the patterns of PV blood flow. In particular, retrograde atrial wave to PV can be a sensitive predictor of elevated LA pressure, and a diastolic wave in PV can be an indicator of LV diastolic dysfunction [9,22]. Also, chronic LV diastolic dysfunction can lead to the anatomical and electrical remodeling of LA [23]. Therefore, it can be deduced that LV diastolic dysfunction might be related to PV remodeling. In the present study, the correlative echocardiography-CCTA analysis demonstrated a positive relationship between PVVI and E/Em; furthermore, PVVI was independently associated with E/Em, indicating that PV enlargement could be directly linked to LV diastolic dysfunction. 

Interestingly, the E/Em-related changing pattern of PVVI differed from the PVVI pattern observed in our study. As E/Em increased, the increase in PVVI was more gradual compared with the increase in LAVI, suggesting that PV is more resistant to structural remodeling than LA. This finding may be attributable to the stretch-resistant physical properties of extra-parenchymal PVs that were previously documented in a human post-mortem study [24]. On the other hand, the curve pattern of PVVI with E/Em showing an exponential nature suggested that the marked PV remodeling begins later than LA in the course of worsening diastolic function. Thus, these findings enable us to postulate that PVVI can be a progress indicator, representing a more severe LV diastolic dysfunction. 

### 4.3. Limitations

Three-dimensional CCTA is a useful imaging tool for the evaluation of PV anatomy and cardiac volume, and its embedded function of automatic calculation facilitates fast and easy assessment [8,25]. However, there is a lack of data regarding the standard methodology for PV volume estimation or sex- and age-specific reference values for the normal range of PV volumes. Therefore, our method requires validation. Although anatomic variation of PV is highly prevalent and recognized as a potential AF substrate, we only assessed the PV volume [26]. Further research evaluating the impact of anatomic variants on total PV volume will be required. In our results, PVVI and E/Em showed a weak correlation. This finding might be because these values were obtained from different imaging modalities at different times. It might also be attributable to the very nature of our data derived from outpatient-based routine echocardiography; the echocardiographic measurements, especially diastolic parameters, could be susceptible to temporal variability [27]. Our study population included a relatively healthy cohort; however, it lacked a sufficient number of patients with severe diastolic dysfunction (E/Em >15; only 10% of patients). This point could lead to a weak statistical power to compare the diastolic parameters between patients with AF and those without AF. A well-designed serial imaging study based on a larger population with a different level of LV diastolic dysfunction is required to validate our findings and investigate other factors responsible for PV enlargement. Finally, the incidence of new-onset AF in our study was relatively low. Even though an effort to search the documented AF patients in detail was made, there might have been subjects with undiagnosed AF. Also, asymptomatic or paroxysmal AF patients often fail to get their arrhythmia recorded or to visit the hospital. Given the study’s retrospective nature and the low incidence of new-onset AF, our findings should be interpreted cautiously.

## 5. Conclusions

The present study provides clinical evidence supporting PV enlargement as a potential consequence of chronically elevated LA and LV filling pressure and could potentially play a causative role in AF development. Further research elucidating the mechanisms behind structural remodeling of PV and its role in AF arrhythmogenesis will help to identify patients at risk, and, consequently, reduce cardiovascular morbidity and mortality.

## Figures and Tables

**Figure 1 jcm-09-00401-f001:**
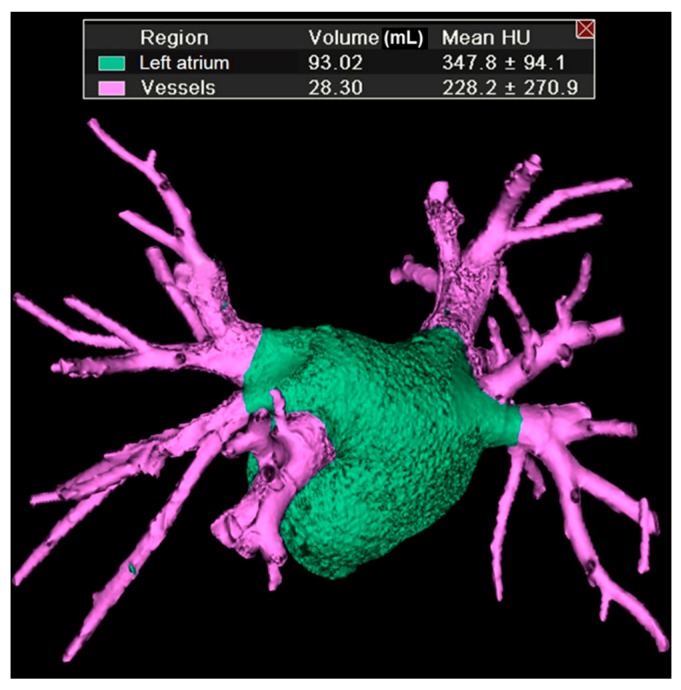
An example of three-dimensional measurements of the pulmonary vein volume using the dedicated image analysis software (Vitrea 2; Vital Images, Minneapolis, MN, USA) on multidetector computed tomography images. HU = hounsfield.

**Figure 2 jcm-09-00401-f002:**
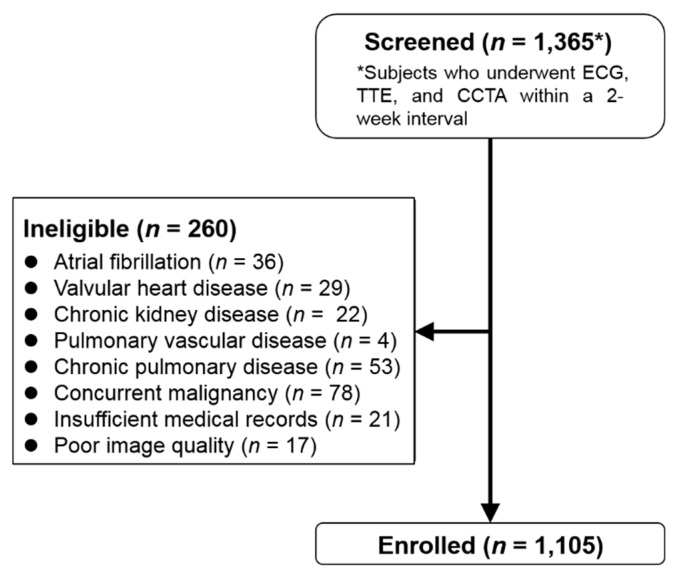
Flow chart for the inclusion and exclusion of the study. CCTA = cardiac computed tomography angiography, ECG = electrocardiography, TTE = transthoracic echocardiography.

**Figure 3 jcm-09-00401-f003:**
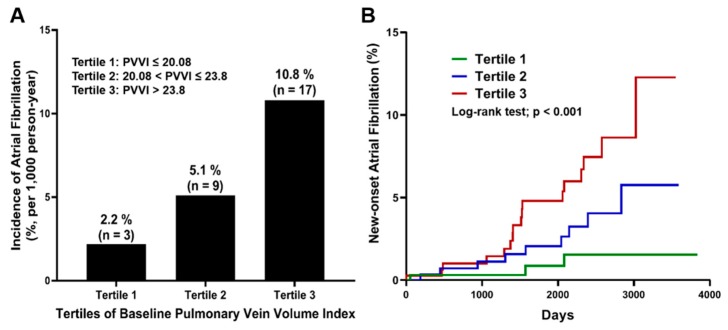
(**A**) Incidence of atrial fibrillation according to the baseline PVVI tertiles. (**B**) Illustration of the cumulative incidence of new-onset AF using the Kaplan-Meyer method. AF = atrial fibrillation, PVVI = pulmonary vein volume index.

**Figure 4 jcm-09-00401-f004:**
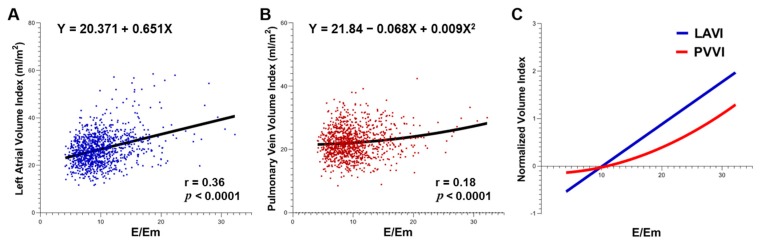
(**A,B**) Correlations of E/Em with LAVI and PVVI, respectively. (**C**) Curve comparison based on normalized LAVI and PVVI values. The difference in E/Em-related changing patterns between LAVI and PVVI is well shown. E/Em = the ratio of the early mitral flow velocity (E) to the early mitral annular tissue velocity (Em), LAVI = left atrial volume index, PVVI = pulmonary vein volume index.

**Figure 5 jcm-09-00401-f005:**
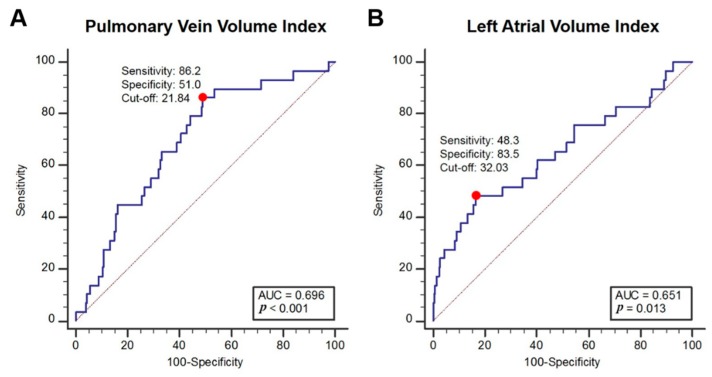
Receiver-operating characteristic curves of PVVI (**A**) and LAVI (**B**), for the prediction of new-onset atrial fibrillation. AUC = area under the curve, LAVI = left atrial volume index, PVVI = pulmonary vein volume index. The red-dots represent the optimal cut-off values of the volume indices.

**Table 1 jcm-09-00401-t001:** Baseline clinical and echocardiographic characteristics according to the new-onset AF status at follow-up.

Variables	Total (*n* = 1105)	AF (*n* = 29)	No AF (*n* = 1076)	*p* Value
Age (year)	58.2 ± 13.4	61.2 ± 13.6	54.6 ± 13.2	0.008
Male (%)	579 (52.4)	12 (41.4)	567 (52.7)	0.261
Body mass index (kg/m^2^)	24.9 ± 3.5	24.9 ± 3.6	24.9 ± 3.6	0.893
Systolic blood pressure (mmHg)	127.5 ± 17.6	122.8 ± 16.6	127.7 ± 17.5	0.153
Diastolic blood pressure (mmHg)	76.6 ± 12.1	76.8 ± 12.0	71.3 ± 12.8	0.021
Pulse pressure (mmHg)	50.9 ± 11.7	50.9 ± 11.6	51.4 ± 16.0	0.807
Hypertension, *n* (%)	465 (42.1)	13 (44.8)	452 (42.0)	0.849
Diabetes, *n* (%)	175 (15.8)	5 (17.2)	170 (15.8)	0.797
Dyslipidemia, *n* (%)	148 (13.4)	5 (17.2)	143 (13.3)	0.577
Coronary artery disease, *n* (%)	229 (20.7)	6 (20.7)	223 (20.7)	1.000
Cerebrovascular accident, *n* (%)	41 (3.7)	0 (0.0)	41 (100.0)	0.622
Heart failure, *n* (%)	23 (2.1)	1 (3.4)	22 (2.0)	0.461
eGFR (mL/min/1.73m^2^)	93.5 ± 25.5	91.9 ± 25.6	93.5 ± 27.5	0.742
β-blocker use (%)	42 (3.8)	1 (3.4)	41 (3.8)	1.000
Calcium channel blocker use (%)	126 (11.4)	4 (13.8)	122 (11.3)	0.564
ACEi/ARB use (%)	312 (28.2)	6 (20.1)	306 (28.4)	0.411
Statin use (%)	122 (11.0)	3 (10.3)	119 (11.1)	1.000
LV ejection fraction (%)	62.0 ± 6.6	63.8 ± 6.8	62.0 ± 6.6	0.152
LV end-diastolic dimension (mm)	47.1 ± 4.9	47.1 ± 4.9	45.4 ± 6.7	0.173
Early mitral flow velocity (E, cm/s)	64.8 ± 17.3	65.6 ± 17.6	64.8 ± 17.3	0.811
Early mitral annular tissue velocity (Em, cm/s)	7.0 ± 2.4	6.2 ± 2.5	7.1 ± 2.4	0.049
E/Em	10.0 ± 3.9	12.4 ± 6.0	10.0 ± 3.8	0.042
Estimated right atrial pressure (mmHg)	5.1 ± 1.0	5.2 ± 1.0	5.1 ± 1.0	0.737
Pulmonary arterial pressure (mmHg)	26.0 ± 5.6	26.0 ± 5.6	26.0 ± 5.4	0.989
LV mass index (g/m^2^)	87.2 ± 21.6	98.0 ± 35.7	86.9 ± 21.0	0.105
LAD (mm) on echo	35.7 ± 4.5	37.9 ± 6.6	35.6 ± 4.4	0.078
LA volume index (LAVI, ml/m^2^) on echo	26.9 ± 7.0	33.4 ± 13.1	26.7 ± 6.7	0.010
PV volume index (PVVI, ml/m^2^) on CT	22.2 ± 4.7	25.3 ± 5.1	22.1 ± 7.5	<0.001

Values are *n* (%) or mean ± SD. ACEi/ARB = angiotensin-converting enzyme inhibitor or angiotensin receptor blocker, AF = atrial fibrillation, CT = computed tomography, eGFR = estimated glomerular filtration rate, LA = left atrial, LAD = LA dimension, LV = left ventricular, PV = pulmonary vein.

**Table 2 jcm-09-00401-t002:** Determinants of pulmonary vein volume index by multiple linear regression analysis.

	B	Standard Error	*β*	*t*	*p* Value
Male	0.760	0.318	0.081	2.389	0.017
LVEDD	0.102	0.033	0.104	3.054	0.002
E/Em	0.127	0.041	0.106	3.130	0.002
LAVI	0.122	0.023	0.182	5.308	<0.001

The variables entered in the model were age, sex, LVEDD, LVMI, LV wall thickness, Em, E/Em, PA pressure, and LAVI. E = early mitral flow velocity, Em = early mitral annular tissue velocity, E/Em = the ratio of E to Em, LAVI = left atrial volume index, LVEDD = left ventricular end-diastolic dimension, LVMI = left ventricular mass index, PA = pulmonary arterial, PVVI = pulmonary vein volume index, LV = left ventricular.

**Table 3 jcm-09-00401-t003:** Cox regression analysis for predicting new-onset atrial fibrillation.

Variable	Univariate Analysis		Multivariate Analysis	
	HR (95% CI)	*p* Value	HR (95% CI)	*p* Value
Age	1.040 (1.010, 1.071)	0.009	1.033 (1.008, 1.058)	0.189
Sex (male)	1.578 (0.746, 3.336)	0.229	1.140 (0.695, 1.665)	0.732
BMI	1.007 (0.907, 1.119)	0.893	0.968 (0.908, 1.040)	0.688
Systolic BP	0.983 (0.960, 1.006)	0.152	0.975 (0.952, 0.992)	0.149
Pulse pressure	1.004 (0.972, 1.037)	0.807	1.021 (0.998, 1.056)	0.361
Hypertension	0.122 (0.534, 2.355)	0.761	0.117 (0.555, 1.581)	0.898
Diabetes mellitus	1.110 (0.418, 2.950)	1.211	1.211 (0.706, 1.826)	0.674
CAD	0.998 (0.401, 2.480)	0.996	0.613 (0.086, 1.180)	0.403
LV mass index	1.016 (1.004, 1.028)	0.008	1.012 (1.002, 1.020)	0.256
Em	0.838 (0.701, 1.001)	0.051	1.145 (0.989, 1.315)	0.386
E/Em	1.105 (1.037, 1.178)	0.002	1.046 (0.967, 1.113)	0.590
PA pressure	1.000 (0.932, 1.074)	0.989	1.021 (0.895, 0.983)	0.152
LAVI on echo >32	3.721 (3.330, 4.112)	0.001	2.754 (2.098, 3.196)	0.076
PVVI on CT >21.8	5.949 (5.410, 6.488)	0.001	5.401 (4.931, 6.193)	0.007

Among common risk factors of developing AF, congestive heart failure was excluded due to its low incidence. Relevant echocardiographic covariates were selected based on the intergroup comparison results. BMI = body mass index, BP = blood pressure, CAD = coronary artery disease, CI = confidence interval, CT = computed tomography, E = early mitral flow velocity, Em = early mitral annular tissue velocity, E/Em = the ratio of E to Em, HR = hazard ratio, LAVI = left atrial volume index, PA = pulmonary arterial, PVVI = pulmonary vein volume index, LV = left ventricular.

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
