# Peer review of "Pulmonary Vein Enlargement as an Independent Predictor for New-Onset Atrial Fibrillation"

_jcm, 2020, doi:10.3390/jcm9020401_

Round 1
Reviewer 1 Report
I read with great interest this retrospective study, in which the authors investigated the role of the pulmonary vein volume index (PVVI) for new-onset atrial fibrillation (AF). In 1105 subjects who underwent transthoracic echocardiography and cardiac CT angiography, the authors showed that increased PVVI was independently associated with new-onset AF. The manuscript is well-written, the statistical analysis is appropriate and the conclusions are supported by the data. Please consider the following comments.
1. How did you define the study period? In addition, how did you define the follow-up period? Please justify.
2. Please provide a flowchart indicating the reasons for the exclusion of the 260 patients.
3. What was the p-value you considered significant in the univariate analysis for entering a parameter in the stepwise regression analysis?
4. Some important data regarding patients’ medications are missing in Table 1. For instance, what was the proportion of patients receiving B-blockers and other antiarrhythmic drugs in each group?
5. Correlations between LAVI and PPVI and E/Em are very weak and these results should be interpreted with great caution. Thus, I would remove this analysis from the manuscript. The manuscript (especially the discussion) should be modified accordingly.
6. Please provide the limits of the intervals for each PVVI tertile.
7. The units of the y-axis are missing in figure 2.
8. The very low incidence of new-onset AF over the four-year follow-up period is unexpected. Please discuss this specific point.
9. What is the mean delay of new-onset AF in your cohort? It is difficult to conceive that a single value of PPVI could result in new-onset AF a few years later. Too many other confounding factors could be responsible for new-onset AF during such a follow-up period. As an illustration, the area of the ROC curve for PVVI is quite small and the mean value of PVVI in patients without new-onset AF is > 21.8. Thus, the results of the multivariate analysis should be considered with caution. This needs to be discussed in more detail.
10. The abbreviations used in the abstract and in the manuscript are unclear. Especially, you mention either PVV or PVVI, which is confusing to readers. Please clarify this point and check that you have correctly defined all the abbreviations used in the manuscript.
Author Response
Responses to Reviewer #1:
The authors thank the reviewer for helpful and constructive comments.
I read with great interest this retrospective study, in which the authors investigated the role of the pulmonary vein volume index (PVVI) for new-onset atrial fibrillation (AF). In 1105 subjects who underwent transthoracic echocardiography and cardiac CT angiography, the authors showed that increased PVVI was independently associated with new-onset AF. The manuscript is well-written, the statistical analysis is appropriate and the conclusions are supported by the data. Please consider the following comments.
How did you define the study period? In addition, how did you define the follow-up period? Please justify.
The authors thank the reviewer for the insightful comment. As CCTA in our hospital was installed in 2002, we defined the study period from January 2002. Also, we set the study period of interest (2002–2012) to ensure a statistically significant volume of study subjects. To ensure clinically meaningful observation time, we defined the follow-up period as from the date of the CCTA scanning to the day of the last follow-up or to December 2017. More details regarding the study design and follow-up period are now described in greater detail in the manuscript (page 2, line 53-55; page 2, line 59-61; page 2, line 66-67).
Please provide a flowchart indicating the reasons for the exclusion of the 260 patients.
Thank the reviewer for the very constructive comment. As your suggestion, a new figure (Figure 2) showing a flowchart of study inclusion and exclusion criteria has been provided as follows (page 4; line 127-129);
Figure 2. Flow chart for the inclusion and exclusion of the study. CCTA = cardiac computed tomography angiography, ECG = electrocardiography, TTE = transthoracic echocardiography.
What was the p-value you considered significant in the univariate analysis for entering a parameter in the stepwise regression analysis?
We performed a multivariate logistic regression analysis to determine predictors of new-onset AF. So, all potential parameters suggestive of an association with newly developed AF by univariate analysis (p<0.1) and conventional AF risk factors, were included in a stepwise regression analysis as the variable-selection process. This point is now clarified in the ‘Statistical Analysis’ section of the manuscript (page 3, line 113-115).
Some important data regarding patients’ medications are missing in Table 1. For instance, what was the proportion of patients receiving B-blockers and other antiarrhythmic drugs in each group?
Thank the reviewer for the very constructive comment. The authors fully agree with the reviewer’s opinion. The information about patients’ medications is essential, because some medicines, including beta-blockers, angiotensin-converting enzyme inhibitors (or angiotensin receptor blockers), and statins, etc., can affect the new-onset atrial fibrillation. Among our study subjects, some participants have been taking the above medications for the treatment of hypertension, angina, or dyslipidemia. Merely none of the study population have been taking other anti-arrhythmic medications (such as class Ic or class III anti-arrhythmic drugs, verapamil, etc.) at baseline, because arrhythmia patients did not included in this study. Data regarding patients’ medications are now reported in the ‘Results’ section, Table 1 (page 4).
Correlations between LAVI and PVVI and E/Em are very weak and these results should be interpreted with great caution. Thus, I would remove this analysis from the manuscript. The manuscript (especially the discussion) should be modified accordingly.
The authors thank the reviewer for pointing to one of the limitations of our study. The authors also agree with the reviewer’s opinion in a degree. The authors thought that the weak relationship observed between PVVI and E/Em was because these values were obtained from different imaging modalities at different times. It might also be attributable to the very nature of our data derived from outpatient-based routine echocardiography. The echocardiographic measurements, especially diastolic parameters, could be susceptible to temporal variability (Gottdiener JS, et al. Journal of the American College of Cardiology, 1995;25:424-30). Please consider our explanation with your sincere generosity. This point is now included in the “Limitations” section of the manuscripts (page 8, line 256-260).
Please provide the limits of the intervals for each PVVI tertile.
As requested by the reviewer, the limits of the intervals for each PVVI tertile are now provided in the ‘Results’ section (page 5, line 142-143).
The units of the y-axis are missing in figure 2.
As the reviewer pointed out, we added the missing units of the Y-axis in Figure 2 as follows (page 5, line 148-150);
The very low incidence of new-onset AF over the four-year follow-up period is unexpected. Please discuss this specific point.
Although we searched the documented AF patients in detail, there might be subjects with undiagnosed AF. Also, asymptomatic or paroxysmal AF patients could often fail to get their arrhythmia recorded or to visit the hospital. These factors may be related to the low incidence of newly developing AF in our study. This point is now described in the ‘Limitations’ section in the manuscript (page 8, line 266; page 9, line 267-268 ).
What is the mean delay of new-onset AF in your cohort? It is difficult to conceive that a single value of PVVI could result in new-onset AF a few years later. Too many other confounding factors could be responsible for new-onset AF during such a follow-up period. As an illustration, the area of the ROC curve for PVVI is quite small and the mean value of PVVI in patients without new-onset AF is > 21.8. Thus, the results of the multivariate analysis should be considered with caution. This needs to be discussed in more detail.
The authors thank the reviewer for constructive and insightful comments. The authors fully agree with the reviewer’s opinion. As described in the manuscript, the mean delay of new-onset AF was about 4.28 years. Because of the absence of the definition of PV enlargement, before univariate and multivariate analyses, we stratified the PVVI variable by a cut-off value determined using ROC analysis. Therefore, the results of the multivariate analysis in our study should be interpreted cautiously. In multivariate analysis, when adjusting all potential risk factors of developing AF (including LA volume), PVVI remained a significant factor for new-onset AF. This finding suggests that PV enlargement might be an independent predictor for developing AF. Please consider our explanation with a sincere generosity.
The abbreviations used in the abstract and in the manuscript are unclear. Especially, you mention either PVV or PVVI, which is confusing to readers. Please clarify this point and check that you have correctly defined all the abbreviations used in the manuscript.
Thank the reviewer for constructive comments. As suggested by the reviewer, to avoid confusion between PVV and PVVI, we have changed all ‘PVV’ used in the manuscript (abbreviation for pulmonary vein volume) into PV volume. Also, we have correctly defined all of the other abbreviations used in the manuscript.

Reviewer 2 Report
This is a retrospective study where the authors tried to highlight on the relationship between PVVI and the new onset of atrial fibrillation.
The study is well designed and the results are good illustrated to the readers.
The relatively large study population is another advantage of this paper.
- How ever I ‘ve consideration to be addressed: Data concerning medical history mainly medications (statines, beta blocker, ACE-inhibitrs ) , renal function (eGFR) are parameters that to be involved in modifying the cardiac/pulmonary These parameters should be included in the study analysis and correlated with the imaging parameters.
- What is the percentage of COPD patients in both AF- and not AF populations. This is important point that is involved in modifying the imaging parameters.
- The E/Em ratio as shown in table one is more near to insignificance (p 0.049): how would the authors interpret this point
Author Response
Responses to Reviewer #2:
The authors thank the reviewer for the helpful and constructive comments.
This is a retrospective study where the authors tried to highlight on the relationship between PVVI and the new onset of atrial fibrillation.
The study is well designed and the results are good illustrated to the readers.
The relatively large study population is another advantage of this paper.
- However, I‘ve considered to be addressed: Data concerning medical history mainly medications (statins, beta-blockers, ACE-inhibitors), renal function (eGFR) are parameters that to be involved in modifying the cardiac/pulmonary. These parameters should be included in the study analysis and correlated with the imaging parameters.
Thank the reviewer for the critical comment. The authors fully agree with the reviewer’s opinion that some medications (including beta-blockers, angiotensin-converting enzyme inhibitors [or angiotensin receptor blockers], and statins, etc.) and renal function, can affect the cardiac/pulmonary remodeling, further newly developing atrial fibrillation. Some missing data regarding patients’ medications and renal function are now added in the ‘Results’ section, Table 1 (page 4).
- What is the percentage of COPD patients in both AF- and not AF populations. This is important point that is involved in modifying the imaging parameters.
The authors thank the reviewer for the much valuable comment. The authors fully agree with the reviewer’s opinion. As the reviewer rightly pointed out, chronic obstructive pulmonary disease (COPD) leads to chronic hypoxemia and further can be involved in the structural alterations of pulmonary vasculature and heart. Also, COPD has been associated with a high incidence of arrhythmias.
Therefore, the authors have initially screened and excluded the patients who had COPD and pulmonary vascular disease when selecting the study subjects. This point is now clarified in the ‘Materials and Methods’ section (page 2, line 58-59) and new Figure 1 (page 4, line 128-130). The authors are very sorry for making the reviewer confused. The authors beg the reviewer’s deep consideration.
- The E/Em ratio as shown in table one is more near to insignificance (p 0.049): how would the authors interpret this point?
In our results, the E/Em ratio in patients with new-onset AF was higher than in patients without AF. However, as the reviewer pointed out, the statistical power tended to be weak (p=0.042). The authors thought it was because our study population included a relatively healthy cohort and lacked a sufficient number of patients with severe diastolic dysfunction. This point is now clarified in the manuscripts (page 8, line 260-263).

Round 2
Reviewer 1 Report
The authors have taken into account all my comments and suggestions and have improgved their manuscript. I have no further comments.
Reviewer 2 Report
The authors gave clear answers to the suggested comments.